# Silicon Nanowire Field-Effect Transistor as Label-Free Detection of Hepatitis B Virus Proteins with Opposite Net Charges

**DOI:** 10.3390/bios11110442

**Published:** 2021-11-10

**Authors:** Suh Kuan Yong, Shang-Kai Shen, Chia-Wei Chiang, Ying-Ya Weng, Ming-Pei Lu, Yuh-Shyong Yang

**Affiliations:** 1Department of Biological Science and Technology, National Yang Ming Chiao Tung University, Hsinchu 300, Taiwan; wendyyongsk.bt05g@nctu.edu.tw; 2Institute of Biomedical Engineering, National Yang Ming Chiao Tung University, Hsinchu 300, Taiwan; fr509.cm05g@nctu.edu.tw (S.-K.S.); chiangwill0813@gmail.com (C.-W.C.); lynneweng8571@gmail.com (Y.-Y.W.); 3Taiwan Semiconductor Research Institute, National Applied Research Laboratories, Hsinchu 300, Taiwan

**Keywords:** hepatitis B virus, hepatitis B virus surface antigen (HBsAg), hepatitis B virus X protein (HBx), chronic hepatitis B (CHB), hepatocellular carcinoma (HCC), biomarker, protein–protein interaction, biosensor, silicon field-effect transistor (SiNWFET), polycrystalline silicon field-effect transistor (pSiNWFET)

## Abstract

The prevalence of hepatitis B virus (HBV) is a global healthcare threat, particularly chronic hepatitis B (CHB) that might lead to hepatocellular carcinoma (HCC) should not be neglected. Although many types of HBV diagnosis detection methods are available, some technical challenges, such as the high cost or lack of practical feasibility, need to be overcome. In this study, the polycrystalline silicon nanowire field-effect transistors (pSiNWFETs) were fabricated through commercial process technology and then chemically functionalized for sensing hepatitis B virus surface antigen (HBsAg) and hepatitis B virus X protein (HBx) at the femto-molar level. These two proteins have been suggested to be related to the HCC development, while the former is also the hallmark for HBV diagnosis, and the latter is an RNA-binding protein. Interestingly, these two proteins carried opposite net charges, which could serve as complementary candidates for evaluating the charge-based sensing mechanism in the pSiNWFET. The measurements on the threshold voltage shifts of pSiNWFETs showed a consistent correspondence to the polarity of the charges on the proteins studied. We believe that this report can pave the way towards developing an approachable tool for biomedical applications.

## 1. Introduction

Hepatitis B virus (HBV) belongs to the Hepadnaviridae family. The 42 nm virion consists of a lipoprotein outer envelope and a nucleocapsid core antigen containing circular, partly double-stranded viral DNA [1]. The viral DNA size is around 3.2 kb with four known genes (e.g., S, C, P, and X) [1,2]. These four genes are overlapping and known as open reading frames (ORF), which S gene, C gene, P gene, and X gene encode for surface protein, core protein and e antigen, polymerase and reverse transcriptase, and x protein, respectively [1]. The infection of HBV causes acute and chronic hepatitis B (CHB) in humans. The number of people with chronic hepatitis B (CHB) condition was estimated at 257 million in 2015 [3]. It is known that CHB patients have a higher risk of developing liver fibrosis, cirrhosis, and hepatocellular carcinoma (HCC) [3]. With the adoption of the 2030 Agenda for Sustainable Development, World Health Organization (WHO) has set a goal to eliminate viral hepatitis by 2030 to reduce chronic hepatitis incidence and annual deaths to 0.9 million and less than 0.5 million, respectively [4].

Hepatitis B virus surface antigen (HBsAg) was discovered in 1965, and it became the hallmark of early diagnosis marker for HBV infection diagnosis [5]. The HBsAg is a glycosylated envelope protein of HBV virion, which composes three parts (e.g., small, medium, and large proteins) [5]. The development of immunoassays that achieved a lower detection limit and the ability to quantify HBsAg commercial kits available has widened interest study on HBsAg [6]. There are two main HBsAg quantification methods, which include Architect HBsAg QT (Abbott Diagnostics) and Elecsys HBsAg II (Roche Diagnostics) [5], and the former is most widely being used. The quantification of HBsAg allows the scientific community to analyze the correlation of intrahepatic HBV DNA and covalently closed circular DNA (cccDNA level) [6]. These findings suggested that HBsAg could be the surrogate marker in monitoring CHB patients for drugs treatment response, viral control status, and prognosis of hepatitis [7]. Besides that, hepatitis B virus X protein (HBx) also has been proven to play a crucial role in hepatocarcinogenesis. Adequate evidence suggested that the HBx acts as a pleiotropic transactivator in many HCC development cellular pathways, including interaction with damage-specific DNA-binding protein 1 [8], centrosomal P4.1-associated protein [9], and nucleic acid metabolism [10]. A recent study also reported that the HBx might bind to the promoter region of DLEU2 long non-coding RNAs, which are frequently dysregulated in HBV-related HCC [11]. These findings showed that HBx could be a potential biomarker for monitoring the pathogenesis of HCC from CHB patients.

HBV vaccination was applied a few decades ago, and the vaccination program has reduced HBV prevalence in many countries. However, in developing countries, HBV prevalence remains high due to the poor prevention, management, and control of the virus infection spreading [12]. The poor virus infection control could be due to many factors such as ignorance, high cost of the vaccination program, lack of healthcare facilities, lack of trained diagnostic personnel, and the treatment protocol extrapolated from developed countries might not be appropriate for developing countries [13]. According to the WHO report, less than 5% of CHB patients know their status, and expensive lifelong diagnostic is unaffordable for most countries [4]. A constant checkup is necessary to monitor CHB and prevent the virus from outspreading during the incubation period. Besides that, the CHB patients might be asymptomatic and cause the virus to spread silently [14]. Thus, it is necessary to develop a simple, low-cost, and ultra-sensitive HBV detection method for sensing key biomarkers, such as HBsAg and HBx, to provide a sustainable diagnosis of HBV infection and early diagnosis of CHB patients.

To date, a broad diversity of field-effect transistor (FET) sensors have been invented and studied for their potential application in the biomedical field [15]. Among these FET sensors, the silicon nanowire field-effect transistor (SiNWFET) sensor shows a label-free sensing principle, and field-effect interaction has attracted significant attention [16]. In addition, the well-developed silicon industry is beneficial to the study of SiNWFET. Many studies have demonstrated the potential of SiNWFET for biosensing applications, including detection of nucleic acids [17], antibodies and antigens [18], and protein–protein interactions [19]. These findings revealed the benefits of SiNWFET, including ultrahigh-sensitivity, real-time, and label-free detection, making it a great potential candidate in biosensor development [20,21]. Generally, an electronic biosensor composed of a bio-receptor and electronic transducer allows for converting the biological information into an electronic signal [16]. Thereby, the direct conversion of a biological reaction to electronic signals has widened the biological applications of the biosensor. Furthermore, the integration of biosensors with an innovative analytic framework could make it act as a good biosensor candidate for point-of-care testing (POCT) and the internet of things (IoT) [22,23,24].

Due to the technical difficulties in the bottom-up fabrication process of SiNWFET, a simple and cost-effective polycrystalline silicon sidewall spacer etching technique has been proposed [25,26,27,28]. This process technique is compatible with the complementary metal-oxide-semiconductor (CMOS) process, which is commonly used in the current commercial semiconductor process. Accordingly, the polycrystalline silicon nanowire field-effect transistors (pSiNWFETs) in this report were fabricated by the commercial process foundry through the sidewall spacer etching technique. Furthermore, the chemical functionalization techniques of the pSiNWFET surface were performed for the immobilization of the relevant antibodies for sensing the HBV-related biomarkers such as HBsAg and HBx proteins. The HBsAg is the hallmark of HBV infection diagnosis and has been discussed in other papers [29,30]. However, HBx protein detection is a new topic that has not been addressed in other articles. Notably, these two proteins possess opposite polarities of the charges to each other, revealing the negative and positive polarities for HBsAg and HBx proteins, respectively. The difference between the threshold voltage of pSiNWFETs before and after the antigen incubation revealed a consistent correspondence to the polarity of the charges on the proteins. By varying the concentrations of the antigens, the equilibrium dissociation constant (*K_D_*) can be extracted, showing the value at the femto-molar level for HBsAg and HBx cases. Our experimental demonstration of using pSiNWFETs as new biosensors to detect the HBsAg and HBx proteins at the femto-molar level provides an opportunity to evaluate and understand hepatitis infection and future biosensor development for the realm of POCT.

## 2. Materials and Methods

### 2.1. Materials

Acetone (99.9%) and ethanol (99.5% and 99.8%) were obtained from ECHO Chemical Co., Ltd. (Miaoli, Taiwan). Glutaraldehyde (GA), 3-aminopropyltrithoxysilane (APTES), sodium cyanoborohydride (NaBH_3_CN), Bis-tris propane, and anti-mouse IgG (whole molecule)–gold antibody (Lot. Number G7652) were obtained from Sigma (St. Louis, MO, USA). Hepatitis B surface antibody (HBsAb, Lot. Number GTX36859), hepatitis B surface antigen (HBsAg, Lot. Number GTX57164), hepatitis B virus X protein antibody (anti-HBx, Lot. Number GTX22741), and hepatitis B virus X protein (HBx, Lot. Number GTX17526-pro) were obtained from Genetex Inc. (Irvine, CA, USA). EKC830 was obtained from DuPont Electronics Technologies, USA.

### 2.2. Device Fabrication

The n-type pSiNWFET were fabricated using a commercial process technology provided by Episil Holding Inc. Taiwan. The pSiNWFET structure comprised two poly-silicon nanowires with 94 nm in width and 2.43 µM in length, which served as conducting channels. The fabrication procedure was performed using the sidewall spacer technique, which has been developed previously [25,26,27].

### 2.3. Device Cleaning, Surface Modification, and Antibody Immobilization

The pSiNWFET was treated with organic solvents, including EKC830 and 99.5% ethanol, to remove the surface of unwanted chemical compounds and the photoresist layer. The EKC830 was heated to 95 °C before pSiNWFET soaked into for 10 min and washed with 99.5% ethanol.

Subsequently, the chemical surface modification for self-assembly of antibodies on pSiNWFET was performed. First, the pSiNWFET was cleaned with plasma cleaner (Harrick Plasma PDC-32G) for 5 min before being immersed into 2% APTES diluted in 99.8% ethanol solution to form a self-assembled monolayer which covalently links between surface silanol groups (SiOH) and terminal with amines groups (NH_2_). Subsequently, the pSiNWFET was cleaned with 99.5% ethanol and heated on a hot plate at 120 °C to remove surplus ethanol. Second, the pSiNWFET was soaked into 2.5% GA mixed in 10 mM Bis-tris propane solution for 30 min, forming a connection of amines group from APTES and a terminal of aldehyde group.

Antibody immobilization was performed by adding 1 µg/mL of HBsAb or 10 µg/mL of anti-HBx that functioned as a probe onto the device surface and incubated for 16 h at 4 °C. The amino acid of the antibody will bind to the aldehyde group of GA. The non-specific binding sides and active amine groups were blocked by 4 mM NaBH_3_CN solution containing 10 mM Tris-HCl buffer (Figure 1). The pSiNWFET was dried with nitrogen gas and kept in a vacuum bag at 4 °C for further experiments.

### 2.4. Surface Modification and Probe Immobilization Verification

X-ray photoelectron spectroscopy (XPS) surface analysis was performed using a PHI Quantera II with an X-ray spot size of 200 nm. This experiment was performed to verify the surface modification step and success of the probe immobilization. The silicon wafer before and after surface modification, as described in Section 2.3 were analyzed its element of carbon (C), nitrogen (N) and oxygen (O) (Appendix A).

A scanning electron microscope (SEM) was used to verify surface modification and probe immobilization on the device. The anti-mouse-gold antibody was used to interact with the immobilized HBsAb. The anti-mouse-gold antibody is an anti-mouse IgG antibody conjugated nano-gold particle with a size range of 10 to 12 nm. The nano-gold particles will be observed using SEM. The anti-mouse-gold antibody was prepared with 0.1% phosphate-buffered saline (1:100) and loaded onto the pSiNWFET before and after pSiNWFET surface modification mentioned in Section 2.3. The anti-mouse-gold antibody was incubated for 2 h at room temperature. Subsequently, the anti-mouse-gold antibody from each pSiNWFET was washed with deionized water and dried with nitrogen gas. The pSiNWFET was coated with a layer of platinum with a power range of 10 to 30 mA for a period range of 10 to 50 s. The pSiNWFET and nano-gold particles were observed under the SEM setting of IST, 10.0 kv, 8.5–8.7 mm, 50.0 k magnification, and SE(U).

### 2.5. Electrical Property of pSiNWFET Measurement

The electrical characterization measurement of pSiNWFET was performed by employing a commercial characterization analyzer (Keithley 2636a). During the electrical measurement of the I_D_–V_G_ curve, the drain voltage was set at a constant value of 0.5 V (V_D_ = 0.5 V), and the gate voltage was swept from −1 to 2 V. To screen the devices with more reliability, we intentionally measured the I_D_–V_G_ curve of each device for three times to check whether there was any evidence of variation over time. The baseline of the I_D_–V_G_ curve was measured under aqueous conditions (Appendix A).

### 2.6. Extraction of Threshold Voltage Shift in pSiNWFET

As aforementioned, the baselines of the electrical properties of pSiNWFETs were measured before loading the analyte samples onto the pSiNWFET surface. Each device was incubated with analyte solution (e.g., HBsAg or HBx) for 30 min. The analyte was prepared in the concentration of 100 fg/mL to 10 pg/mL in 10 mM of Bis-tris propane (pH 7). Then, it was followed by performing a washing process of the device surface with 10 mM of Bis-tris propane buffer. Later, the electrical property measurement was performed, as described in Figure 1. The difference between pSiNWFET threshold voltage before and after analyte incubation was extracted as the sensing signal. The threshold voltage (V_th_) was determined with the intercept of the linear approximation of the I_D_–V_G_ curve at the maximum slope point, as shown in Appendix A [31]. Subsequently, the threshold voltage changing (ΔV_th_) was obtained which the value of analyte V_th_ subtracts the value of baseline V_th_.

### 2.7. Extraction of the Equilibrium Dissociation Constant (K_D_) in pSiNWFET

The data collected from Section 2.6 was further evaluated for obtaining *K_D_* of HBsAg and HBx in pSiNWFET using the following equation [32]:(1)$$\frac{∆I_{ds}}{g_{m}}=∆V_{T}=\frac{q_{A}}{C_{0}}{\left[B\right]}_{max}\times \frac{\left[A\right]}{\left[A\right]+K_{D}}$$
where *q_A_* is the absorbed analytes electric charge, *C*_0_ is the analyte/channel capacitive coupling, [*A*] is the analyte concentration in bulk solution, and [*B*]*_max_* is the maximum surface density of functional binding sites on silicon nanowire [32].

### 2.8. Protein Zeta Potential Measurement

The zeta potential was performed using Zeta-potential & Particle size Analyzer ELSZ-2000 series (Qtsuka Electronics Co., Ltd.) and Zetasizer Nano ZS90 instrument (Malvern Instruments Ltd., United Kingdom). The proteins HBsAg and HBx were individually prepared in 10 mM Bis-tris propane, pH 7, at the concentration of 6.25 µg/mL. The refractive index, dielectric constant, and viscosity of the instrument were set at 1.5650, 78.3, and 0.8878 cP at 25 °C, respectively. The zeta potential of the protein was measured thrice, and the average value was calculated.

## 3. Results and Discussion

### 3.1. Device Structure

Figure 1A shows the schematic of the structure of the pSiNWFET (not in scale). The structure includes a silicon substrate, a stacked silicon nitride/silicon oxide layer, two highly doped Si areas, and two poly-Silicon nanowires (pSiNWs) acts as the back-gate electrode, the gate dielectric, source/drain area, and the conducting channels, respectively. The top-view SEM image of the device is demonstrated in Figure 1B, revealing that two nanowires (NWs) are located on the surface of the nitride layer. The length and width of the NWs are measured to be 2.43 µM and 94 nm, respectively. Besides that, the source and drain are covered with the passivation layer to prevent direct contact with the aqueous and cause a short circuit (Appendix A).

There are two kinds of technical approaches, top-down and bottom-up, for the semiconductor process fabrication of SiNWFET. The top-down approach involves using advanced lithography and reactive ion etching techniques to fabricate the NW structures on the surface of the wafer. In contrast, the bottom-up approach is based on the assembling technique of the NWs grown by the vapor-liquid-solid method to place the NWs on the wafer surface [33]. Although the top-down fabrication approach is amenable to mass production, reliable and reproducible, and has no integration issues. However, it is costly, limited in NW dimension and limited choice of NW material, and time-consuming [34]. The previous studies showed a simple and cost-effective top-down fabrication using the sidewall spacer etching technique [25,26,27,28] that was transferred to a commercial foundry via an industry-academia collaboration cooperation project. The pSiNWFET devices used in this study supplied by the commercial semiconductor company showed that the fabrication technique of pSiNWFET was successfully transferred from an academic laboratory (National Applied Research Laboratories, Taiwan Semiconductor Research Institute) to a commercial foundry. Appendix A shows the electrical characteristics measured from nine devices, revealing the excellent and reliable features. It indicates the mass production of the pSiNWFET by the commercial foundry is highly feasible.

### 3.2. Surface Modification and Probe Immobilization Verification

Surface modification and immobilization of antibodies on the devices is the first critical step that needs to be achieved for developing a biosensor. In this study, the surface modification and self-assembly antibodies on the device with amine and aldehyde linkers were adapted from the previous study [35]. The verification was analyzed using XPS (Figure 2), which XPS is a powerful tool used to analyze surface chemistry [36]. The four surface modifications on silicon wafer include nude, APTES modified, APTES + glutaraldehyde modified (APTES + GA) APTES + glutaraldehyde + HBsAb modified (APTES + GA + HBsAb) were analyzed, as shown in Figure 2. The XPS C1s spectrum showed a peak at 283.5 eV for the nude surface (Figure 2A). Following the surface modification steps, an increased intensity major peak has been observed at the binding energy range of 283.5 eV to 284.5 eV, which may be attributed to C-C/C-H. The increased intensity of the major peak suggests that more carbon elements were found on the surface, which supported the longer linkers immobilized along with the immobilization steps. Other than the nude silicon wafer, minor peaks were observed at the binding energy range of 287.5 eV to 288.5 eV after APTES immobilization, which may be attributed to C-N/C-O/C=O [37,38]. Figure 2B shows the XPS spectrum of nitrogen element of four surface modifications as aforementioned. The peaks observed at 399 eV show increased intensities after being modified with APTES + GA + HBsAb. This result suggested more nitrogen elements were found on the surface and supported longer linkers immobilized on the surface, particularly APTES and antibodies containing nitrogen elements. Similarly, the oxygen element of four surface modifications was analyzed at the binding energy at 532 eV (Appendix A).

Furthermore, we used the SEM tool to verify the validity of the surface modification on devices. The anti-mouse IgG-gold antibody (AuNPs) was used to confirm the success of probe immobilization on pSiNWFET by binding on the immobilized HBsAb, as shown in Appendix A. Appendix A showed the SEM analysis of the nude device (Appendix A) and APTES + glutaraldehyde + HBsAb modified device (Appendix A). The AuNPs specifically bind to the HBsAb but cannot be linked to the nude pSiNWFET. Appendix A showed increased quantities of nano-gold particles that have been observed on functionalized pSiNWFET. These results showed that only the devices with immobilized antibodies observed expected nano-gold particles and indicates that the probes were successfully immobilized onto the pSiNWFET.

### 3.3. Electrical Properties Measurement of Surface Modification and Probe Immobilization

The procedure of probe immobilization was confirmed again by measuring the electrical properties changes, which has been reported in a previous study [35]. As shown in Figure 3, the electrical properties of pSiNWFET were measured following surface modification steps. Figure 3 showed nude device electrical property (G1, black line) and served as the baseline of the device. Then, APTES modified device was measured (G2, red line). The increased I_D_ and decrease in threshold voltage were observed when there is excess positive charge available on the surface for an n-type pSiNWFET. The excess positive charge was contributed by the amine group of APTES (pKa = 4.0) at pH 7. Subsequently, the APTES + glutaraldehyde modified pSiNWFET showed a decrease in I_D_ (G3, blue line), which was caused by the imide bonds formation of the glutaraldehyde, where the positive charge of the amine group of APTES was converted to neutral by the imide bonds. Lastly, the immobilized HBsAb (G4, green line) showed a decrease in I_D_ indicated excess negative charges applied onto the device. The inset figure represents the change in threshold voltage following each surface modification step. This result is consistent with our previous studies [19,35], which determine the surface modification process by measuring the electrical properties of pSiNWFET.

### 3.4. Biosensing of Various Concentrations of HBsAg and HBx

To probe the polarity of HBsAg and HBx, the zeta potential was measured. The zeta potential of HBsAg and HBx in 10 mM BTP (in pH 7) was −9.00 mV and 7.653 mV, respectively. The results revealed that HBsAg and HBx proteins possessed negative and positive polarities, respectively. This result is consistent with studies that showed the isoelectric point (pI) of HBsAg [39] and HBx [40] were 4.6 and 8.3, respectively. It is known that when the pH value is greater than pI, the protein surface is negatively charged or vice versa [41]. Hence, the pH 7 buffer that has been used in this system would cause HBsAg to carry a negative charge, whereas HBx carries a positive charge.

Figure 4 shows the electrical properties of functionalized pSiNWFET response on the various concentration of HBsAg and HBx. Figure 4A shows the biosensing of HBsAg using an HBsAb-immobilized pSiNWFET. The electrical property of a pSiNWFET was conducted at a fixed drain voltage (V_D_ = 0.5 V) and gate voltage sweeping from 0.8 V to 2.0 V. Firstly, the baseline of the pSiNWFET was measured and revealed in black line (G1). Subsequently, the concentration of 100 fg/mL of HBsAg was loaded onto the device and incubated for 30 min. The analyte was removed and replaced with the 10 mM Bis-tris propane on the device. The pSiNWFET showed a decrease in I_D_ and resulted in a positive shift in the threshold voltage (red line, G2). Later, the higher concentration of HBsAg (1 pg/mL or 10 pg/mL) was repeated for the above-mentioned steps. The decreased I_D_ trend was obtained for 1 pg/mL (blue line, G3) and 10 pg/mL (green line, G4) of HBsAg compared to baseline. The normalized value of each sample group was calculated, and the average of 3 devices was presented in the inset figure. The threshold voltage (Appendix A) and the value of threshold voltage changing (ΔV_th_) were calculated. The normalized value of G2–G1 was 120.262 mV, and an increasing trend was observed for G3–G1 (330.728 mV) and G4–G1 (432.247 mV).

Similarly, Figure 4B showed the electrical property of the anti-HBx-immobilized pSiNWFET in biosensing of HBx. The test was conducted at a fixed drain voltage (V_D_ = 0.5 V), and gate voltage sweeps from 0.2 V to 2.0 V. The black line indicates the baseline (G1) of the pSiNWFET, whereas the red, blue, and green lines indicated the electrical property of pSiNWFET after incubating with 100 fg/mL HBx (G2), 1 pg/mL HBx (G3), and 10 pg/mL HBx (G4), respectively. The inset figure presented ΔV_th_ of the average value of three devices after each sample incubation. The normalized value of G2–G1 was −84.005 mV, then the value decreased to −278.552 mV for G3–G1 and to −479.085 mV for G4–G1.

Figure 4 demonstrated that the n-type pSiNWFETs have an outstanding and consistent response to negative-charge and positive-charge proteins. The increasing concentration of HBsAg resulted in a decreased I_D_ trend compared to baseline shows that it is a negatively charged protein. Without ambiguity, the pSiNWFET showed a contrary response to HBx biosensing. An increased concentration of HBx showed an increasing I_D_ trend indicates a positively charged protein. According to the general sensing mechanism of SiNWFET, the antigen–antibody interaction on the surface of the sensing area causes the channel conductance to change according to the charge accumulation. In other terms, for an n-type SiNWFET, when negatively charged antigen binds to the antibody immobilized on the sensor surface, it introduces an accumulation of negative charges on the pSiNWFET surface and subsequently decreases the drain current. On the contrary, the accumulation of positive charges on the pSiNWFET surface increased the magnitude of the drain current [23]. The results are consistent with the zeta potential measurement of the analyte proteins.

Furthermore, the pSiNWFET demonstrated its ultrahigh-sensitive properties in the biosensing of HBV-related proteins. Radioimmunoassay (RIA) or enzyme immunoassays (EIA) are the general serological method to determine HBV infection [6]. This study demonstrates the detection of the lowest concentration of HBsAg that can be detected using pSiNWFET was 100 fg/mL. This sensitivity value is comparable to the commercialized fully automated assays invented by Abbott Diagnostic and Roche Diagnostic, which have the lowest detection limit at 0.2 ng/mL [6]. The higher sensitivity achieved by a SiNWFET sensor is due to the structure of silicon nanowire (SiNW). The SiNW 1-dimensional (1-D) nanomaterial has a high surface-area-to-volume ratio that improves analytical sensitivity in bio-molecules detection. This is because the analytes binding onto the surface of SiNW will cause a bulky depletion or accumulation of carrier of nanometer diameter structure and increase the sensitivity to femto-molar detection limits [34]. In addition, our pSiNWFET was able to detect HBx at the lowest concentration of 100 fg/mL, which was comparable to EIA with a detection limit at the sub-nano range [42]. The sensing principle of SiNWFET allows for the direct conversion of bio-molecule events on the sensing surface to the detectable electrical signal without requiring amplification and provides a direct detection method of protein–protein interaction.

### 3.5. Equilibrium Dissociation Constant (K_D_) of Protein–Protein Interaction on pSiNWFET

In addition to the biosensing of biomolecules using pSiNWFET, the threshold voltage shift upon various concentrations of the analytes could be used to extract the equilibrium constant associated with the behavior of binding and desorption of the protein–protein interaction on the device surface. As shown in Figure 5, the red line indicates the threshold voltage increase accordingly to the increased HBsAg concentration, whereas the blue line indicates the threshold voltage decreases accordingly to the increased HBx concentration. Furthermore, the extraction of the equilibrium dissociation constant (*K_D_*) was obtained by applying Equation (1) in the aforementioned. Accordingly, the values of the extracted *K_D_* for HBsAg-HBsAb, and HBx-anti-HBx in this study were approximately 12 fM and 40 fM, respectively (Figure 5). In the past, the powerful tool to determine *K_D_* was surface plasmon resonance (SPR), but the integration of SPR requires expensive optical components that have limited its application [32]. Recently, the use of label-free SiNWFET platforms to determine the equilibrium constant describing the behavior of the proteins affinity kinetics has been demonstrated [32], revealing the equilibrium constant of fM levels for the protein interactions, which is in agreement with our findings.

Taking the amount of the contributed electric charge for each HBsAg and HBx protein as high as one charge, the surface density of binding sites on the surface of pSiNWFET can be approximately estimated using Equation (1). The *C*_0_ is the planar capacitance featuring the stacked silicon nitride (50 nm in thickness)/silicon oxide layer (85 nm in thickness). As a result, the maximum surface density *[B]_max_* was extracted to be at the level of ~10^13^/cm^2^ for the HBsAg and HBx proteins, consistent with the typical values reported in the literature [43,44].

### 3.6. Prospective of pSiNWFET Sensor as Multiplexing Biochip for HBV Detection in Monitoring Patients’ Status

To date, various types of immunosensors have been proposed for HBV infection diagnosis [45], including HBV X gene detection [46], and HBsAg detection [29,30]. These sensors were developed to address the disadvantages of current diagnosis methods, such as the required bulky instruments or complicated procedures. In other words, the ultimate goal is to develop a reliable and feasible tool for simple, sustainable, and cost-effectively diagnosis of HBV infection and the long-term monitoring of CHB patients. A list of other FET-based sensors related to HBV detection has been presented in Table 1. In this paper, the CMOS-compatible fabrication technique was adopted to fabricate the pSiNWFETs [25,28], which was successfully demonstrated in a commercial process foundry in Taiwan. It is showing a great opportunity of commercially using the pSiNWFET for HBV-related proteins biosensing applications. Our pSiNWFET demonstrated its capability in detecting the femto-range concentration of target proteins, which is comparable to other FET-based sensors (Table 1). The lower concentration detection of HBV-related proteins or biomarkers could positively impact hepatitis diagnosis, particularly monitoring the titer of CHB-related biomarkers. In addition, the achievement of the functionalization technique and maturity of the semiconductor industry provides an opportunity to develop a multiplexing HBV biomarkers detection biochip. The biochip could be immobilized with multiple types of HBV infection diagnosis biomarkers and provide high-throughput biosensing.

As aforementioned in Section 3.4, the detection mechanism of SiNWFET response to the analytes was based on the sensor surface depletion or accumulation of the charge. In contrast to nucleotide detection that detects a negative carrier using NWFET, the uncertainty of the analyte protein net charge towards a pH range would affect the interpretation of the sensor response in the protein detection [47]. In this paper, the pSiNWFET demonstrated can differentiate two opposite net charges of the proteins in a given pH 7 by evaluating the electrical properties response. These results revealed a possibility of using pSiNWFET in determining the unknown net charge of a protein via evaluating the electrical properties.

Although early diagnosis or monitoring of hepatitis infection could be achieved by simple, ultrahigh-sensitive, and affordable pSiNWFET. The pSiNWFET also faced a few challenges, including the robustness of the devices towards detecting the real sample. Real samples, such as serum or whole blood, would cause inconsistent results due to non-specific proteins adsorption onto the sensing area and eventually causing noise to the sensor. This could be overcome by optimizing the blocking step of surface modification or by applying anti-fouling pre-treatment on the sensor. Besides that, the washing step commonly used in EIA could be adopted in the biosensing procedure to remove non-covalent binding on the non-specific protein adsorption. However, the efficiency of adopting the washing step has yet to be identified.

## 4. Conclusions

In this paper, we have used the pSiNWFET biosensor fabricated by the commercial process foundry for specific, ultrahigh-sensitive, and label-free detection of HBV-related proteins, namely HBsAg and HBx. In addition to measuring the electrical properties of the pSiNWFETs to confirm the stability and reliability, the validity of the chemical functionalization process of the pSiNWFETs was confirmed using XPS and SEM. Furthermore, the functionalized pSiNWFETs were used to detect two biomarkers, HBsAg and HBx, that carried negative and positive charges, respectively, which were determined by the zeta potential measurement. The experimental results of the threshold voltage shift in pSiNWFETs consistently reflect the charge polarity of the protein targets. In addition, we suggested that pSiNWFET could be (1) a potential candidate for multiplexing detection of HBV-related biomarkers; (2) the electrical noise could be reduced by optimizing the blocking step of functionalizing or adapting the washing step. We expect this approach to positively impact hepatitis B infection diagnosis and widen the study of HBsAg and HBx concentration in HCC development.

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
