# Peer review of "Silicon Nanowire Field-Effect Transistor as Label-Free Detection of Hepatitis B Virus Proteins with Opposite Net Charges"

_biosensors, 2021, doi:10.3390/bios11110442_

Round 1
Reviewer 1 Report
The manuscript by Yong et al. describing a label-free detection of HBV proteins with opposite net charges by SiNWFET may provide useful information and attract the researchers in this field, if the following points are elucidated and carefully revised.
1. The idea of employing FETs (and other kinds of biosensors) for HBV detection is no longer novel and there are several publications in this topic. What is the originality/novelty of this manuscript in comparison with them? A table summarizing the achievements in this field and highlighting its originality/novelty is necessary for the readers to follow.
2. In Figure 2 depicting XPS data, Carbon cannot be used as the factor to verify the surface modification because a layer of adventitious Carbon can be formed as long as the samples are exposed to the air. Instead, Nitrogen, which is only available in APTES and proteins, should be the decisive element. Therefore, Figure S3A should also be placed in the main manuscript for discussion.
3. Control experiments should be carried out to guarantee that the signal changes were from the specific binding between probes and targets. It is also necessary to complement the manuscript with the important information about the sequences of probes and targets.
4. What are the quantification ranges and the cut-off values for detection of HBsAg and HBx in diagnosis and prognosis? Are these ranges with only three concentrations (100 fg/mL – 10 pg/mL) enough to conclude that early diagnosis or monitor hepatitis prognosis could be achieved by SiNWFETs, as claimed in the manuscript?
5. The manuscript will be more convinced if Figure S4 can present the gold nanoparticles with positions on the nanowires.
6. How was the maximum surface density of functional binding sites on silicon nanowire ([B]max in formula 1) calculated? Can the sensors developed in this study detect these 2 targets at the concentrations higher than 10 pg/mL?
Some other minor points:
7. Title: should it be “opposite net charges”?
8. Line 146: 2.5% GA but in Figure 1 and S3: 12.5% GA.
9. Line 248-249: which the XPS which is the powerful tool …
Reviewer 2 Report
Reviewer: In the paper titled "Silicon nanowire field-effect transistor as a label-free detection of hepatitis B virus proteins with net opposite charges" was considered carefully. This topic is worthy of research and the author has made an in-depth analysis and obtained satisfactory results. The data and charts are relatively sufficient, which makes the paper meet the journal standards. Therefore, this article may be suitable for publication after minor revisions. Some important comments:
- The introduction part describes too much background, and the first and third paragraphs are repeated.
- Sections 2.4 and 2.5 lack the citations of supplementary materials.
- It is recommended that the content of Section 2.3 be divided into paragraphs according to sub-headings.
- The author indicated that the sensor has been successfully constructed and the sensitive detection of markers has been achieved, but the linear equation has not been found in the manuscript.
- The English in the manuscript needs to be polished.
Reviewer 3 Report
This manuscript reports on the “Silicon nanowire field-effect transistor as a label-free detection 2 of hepatitis B virus proteins with net opposite charges.” The content of the work is interesting, but the manuscript cannot be published in the present form due to the following issues:
- The author should bring down the plagiarism of the manuscript presently it's around 23%.
- For the proper understanding of the device, FIB-TEM is required for the whole pSiNWFET structure.
- The addition regarding the surface roughness is an important aspect for understanding the behavior of the material, therefore, AFM of the two pSiNWs structure includes the nitride layer of the pSiNWFET structure is mandatory.
- The author explains more regarding the reliability of the pSiNWFETs structure which has been claimed to be done through the electrical property explanation of pSiNWFETs.
- For the understanding of the functionalization carried out during the fabrication of pSiNWFETs structure FTIR is required at every step of functionalization.
- Future aspects of the research should be pointwise mention in the conclusion part
- Careless typological errors have been found in Figure 2.
- Schemes come under the count of Figures so fix the Figure nos accordingly
- Language needs substantial improvement. Please consult a native English speaker or a language editing service.
Round 2
Reviewer 1 Report
The manuscript can be accepted since most of its important points have been properly revised.
Reviewer 3 Report
As the author(s) have replied to all the comments properly therefore manuscript can be accepted.